# Investigation into Efficacy and Mechanisms of Neuroprotection of Ashwagandha Root Extract and Water-Soluble Coenzyme Q10 in a Transgenic Mouse Model of Alzheimer’s Disease

**DOI:** 10.3390/nu17162701

**Published:** 2025-08-20

**Authors:** Caleb Vegh, Gabrielle Walach, Keanna Dube, Bromleigh Dobson, Rohan Talukdar, Darcy Wear, Hasana Jayawardena, Kaitlyn Dufault, Lauren Culmone, Subidsa Srikantha, Iva Okaj, Rachel Huggard, Jerome Cohen, Siyaram Pandey

**Affiliations:** 1Department of Chemistry & Biochemistry, University of Windsor, 401 Sunset Avenue, Windsor, ON N9B 3P4, Canada; veghc@uwindsor.ca (C.V.); d.wear@mail.utoronto.ca (D.W.); jayawarh@uwindsor.ca (H.J.); dufau@uwindsor.ca (K.D.); culmonel@uwindsor.ca (L.C.); srikants@uwindsor.ca (S.S.); okaji@uwindsor.ca (I.O.); huggard1@uwindsor.ca (R.H.); 2Department of Integrative Biology, University of Windsor, 401 Sunset Avenue, Windsor, ON N9B 3P4, Canada; walachg@uwindsor.ca (G.W.); dube112@uwindsor.ca (K.D.); dobson41@uwindsor.ca (B.D.); talukda1@uwindsor.ca (R.T.); 3Department of Psychology, University of Windsor, 401 Sunset Avenue, Windsor, ON N9B 3P4, Canada

**Keywords:** Alzheimer’s disease, amyloid-β, ashwagandha, astrocyte, autophagy, inflammation, oxidative stress, Tonic

## Abstract

Background: Alzheimer’s Disease (AD) is one of the most prevalent neurodegenerative disorders and the most common form of dementia. Although current treatments examine disease progression, many have side effects and primarily target symptomatic relief as opposed to halting further neurodegeneration. Objective: The current study aims to determine the neuroprotective effects of water-soluble coenzyme Q10 (Ubisol-Q10) and an ethanolic Ashwagandha extract (E-ASH) on a transgenic mouse model of AD. Methods: A variety of immunofluorescence staining of biomarkers was conducted to assess mechanisms commonly implicated in the disease. Additionally, spatial and non-spatial memory tests evaluated cognitive functions at two timepoints throughout the progression of the disease. Results: A substantial reduction in microglial activation and amyloid-β (Aβ) plaques when treated with a combination of natural health products (NHPs), Ubisol-Q10 and E-ASH. Moreover, activation of autophagy was upregulated in both the Ubisol-Q10 and combination (Ubisol-Q10+E-ASH given as a combined “Tonic” solution) groups. Oxidative stress was decreased across treated groups, while astrocyte activation was elevated in both the E-ASH and Tonic group. The Tonic group expressed an elevation in the fluorescent intensity of neuronal nuclei (NeuN) and brain-derived neurotrophic factor (BDNF) levels. Interestingly, treatment with E-ASH and Ubisol-Q10 enhanced synaptic vesicle formation compared to controls. Pre-mortem memory tests revealed the treatments to be effective at preserving cognitive abilities. Conclusions: Based on these findings, the combination of E-ASH and Ubisol-Q10 may effectively mitigate the various mechanisms implicated in AD and ultimately prevent further disease progression.

## 1. Introduction

Alzheimer’s disease (AD) is the most common neurodegenerative disease and the leading form of dementia around the world. AD is characterized by progressive loss of neuronal density and connectivity [1,2,3,4,5] in the hippocampus and cortex, leading to cognitive decline, memory loss, morbidity, and death [6]. The core cellular and molecular disruptions associated with AD do not occur in isolation, and instead coincide with symptomatic onset [7,8]. In moderate stages of AD degeneration, hallmark symptoms of memory loss begin to become obvious [9,10]. Various properties of AD progression are believed to be responsible for such declines in cognitive abilities. Key pathological features of AD include the accumulation and deposition of amyloid-β (Aβ) fibrils, forming extracellular neuritic plaques, and the aggregation of hyperphosphorylated tau proteins that form neurofibrillary tangles [11,12,13,14,15,16]. Other biochemical features of the disease include mitochondrial destabilization [17,18,19,20], generation of reactive oxygen species (ROS) [19,21,22,23,24,25], inhibition of autophagy [24,26,27], and neuroinflammation [12,22,28]. Microglia and astrocytes play a significant role in the neuroinflammation associated with AD [12,22,28]. Microglia become activated in the presence of Aβ plaques [29,30] (microgliosis), where they transition from their ramified to ameboid state [28,31,32]. In this state, microglia release various pro-inflammatory cytokines [33], increasing neuronal stress in the cells surrounding Aβ plaques and potentially triggering apoptosis [32,34,35]. These glial cells are known to communicate to either activate or inactivate one another [36,37], which further exacerbates the neuroinflammation in AD [38]. As a result of these multiple mechanisms and key players, AD should be considered a multi-factorial disease, and thus, treatments targeting only one mechanism may prove futile in preventing the characteristic neurodegeneration of the disease.

Current common treatment options for AD include the use of acetylcholinesterase inhibitors (AchEI) and memantine, a NMDAR antagonist [39,40,41,42,43]. AChEIs show primarily symptomatic relief, without addressing pathogenesis [40,42,43] and ultimately failing to reverse or halt AD neurodegeneration [39,41]. Other recently approved treatments include monoclonal antibodies that target Aβ, however concerns regarding cost [44] as well as the efficacy and safety of long-term use [45] have been raised. To address a number of these concerns, we suggest the use of multi-targeting natural health products (NHPs) in the treatment of AD. A multifaceted approach to treatment is preferable, which could be achieved with the use of NHPs, as they have the potential to target a variety of biochemical mechanisms in a single dose. Various disciplines of traditional medicine have used NHPs for thousands of years to address several health concerns [46,47]. NHPs are well-tolerated, safe for long-term use, and easily accessible, which addresses limitations with current therapies and suggests they may be a preferable treatment option for those with AD.

The current study evaluates two NHPs in the treatment of AD: water-solubilized coenzyme Q10 (Ubisol-Q10, patented by Micillic Lda, Lisbon, Portugal) and ethanolic Ashwagandha (*Withania somnifera*) root extract (E-ASH). Ubisol-Q10 has been shown to reduce oxidative stress, improve long-term and working memory, and drastically inhibit Aβ plaque formation in 16-month-old transgenic mice displaying AD-type pathology [22]. In addition, Ubisol-Q10 protects against neuronal cell death by stabilizing mitochondrial functions and inhibiting oxidative stress in in-vitro studies involving human fibroblasts from AD patients [19,22,48,49,50]. While Ubisol-Q10 effectively treats most AD pathologies in transgenic mice, it does not target all mechanisms (ex. Aβ plaques are reduced, but not completely eliminated [22]). Ashwagandha (ASH) has been used to promote brain health in traditional schools of Indian medicine, including *Ayurveda* [51]. It exhibits neuroprotective properties in in-vitro models of AD [52,53], and has been shown to reduce the amount of pro-inflammatory cytokines and modulate the stress response [54,55]. Beyond cellular benefits, previous research has indicated that both Ubisol-Q10 and ASH have the potential to improve memory performance in transgenic mice [22,53]. Since E-ASH and Ubisol-Q10 target various mechanisms, such as oxidative stress and inflammation, their combined use in the Tonic solution could provide a more effective therapy for AD.

In this study, we investigate the efficacy of Ubisol-Q10 in combination with E-ASH in ameliorating progression of AD pathologies, including reducing Aβ plaques, neuroinflammation, and cognitive deterioration. We aimed to elucidate the neuroprotective mechanisms of these treatments by assessing their effects on inflammation, oxidative stress, autophagy, astrocyte activation, and synaptic health. In addition to analysis of biochemical features of AD pathology, confirmation of the treatment’s protection of pre-mortem cognitive AD-degeneration is investigated using the Novel Location Recognition (NLR) and Novel Object Recognition (NOR) tests in an X-maze. The current study builds on previous work from our lab [22] by introducing the anti-amyloid and pro-neuroprotective properties of E-ASH. With this novel Tonic treatment, we aim to ameliorate a vast array of AD pathologies. Our multi-disciplinary approach allows for full-spectrum analysis of Alzheimer’s-type neurodegeneration and evaluation of the efficacy of treatment via Ubisol-Q10 and E-ASH.

## 2. Materials and Methods

### 2.1. Extraction of Ashwagandha Root and Phytochemical Assessment

Ashwagandha root powder (obtained from Premier Herbal Inc., North York, ON, Canada) was extracted using anhydrous ethanol as described previously by Vegh et al. (2021) [56]. The crude extract was filtered through a P8 paper filter, and ethanol was removed using a rotary evaporator. The solid extract was resuspended at 200 mg/mL with anhydrous ethanol. The resuspended extract was analyzed using ultra-performance liquid chromatography coupled with ultra-violet spectroscopy to assess withanolide content and colorimetric analysis based on Dowd’s reagent for flavonoid content. Phytochemical analyses were performed by Laboratoire PhytoChemia (Chicoutimi, QC, Canada). Report of the extract contents is shown in Table 1, which is directly referenced from Vegh et al. (2021) [56].

### 2.2. Animal Care Treatment Regimen

Transgenic AD mice, genetically modified to include human gene mutations associated with AD, allow for the evaluation of pre-mortem behavioural and cognitive changes, and post-mortem biochemical mechanistic changes. To assess cognitive changes, and the potential of the Tonic (Ubisol-Q10 formulation patented by Micillic Lda, Lisbon, Portugal) treatment in protecting memory, a moderately progressing double transgenic mouse model is utilized.

The animal care, treatments, and procedures used in the study were approved by the University of Windsor’s Animal Care Committee following the Canadian Council for Animal Care guidelines (Animal Utilization Project Protocol #17-14). A total of 36 male mice were used, 29 of which were double transgenic, expressing both chimeric mouse/human amyloid precursor protein (Mo/HuAPP695swe) and mutant human Presenilin-1 (PSEN1-dE9) transgenes [57] (Jackson Laboratory; Strain: B6C3-Tg (APPswer,PSEN1dE9)85Dbo/Mmjax; stock #34832). This mouse model brings insight into how humans would progress with early-onset Alzheimer’s disease, and therefore, Aβ plaque formation begins at approximately 6 months of age, continuing its progressive accumulation until 12 months of age. The mice arrived at approximately 2 months of age; however, two transgenic mice deceased prematurely, unrelated to experimental factors, resulting in the exclusion of their data from the study. Additionally, seven male C57BL/6 wild-type mice (Charles River Laboratories) were used. The mice were received at one month of age and were then acclimated to the lab environment. This acclimation process occurred over a two-week period prior to the start of any treatment.

The transgenic mice were randomly assigned to one of four possible treatment groups; the untreated group received normal drinking water (*n* = 6), while the treated group received supplemented water with Ubisol-Q10 (50 μg/mL; *n* = 6), E-ASH (2 mg/mL; *n* = 8), or Tonic (50 μg/mL Ubisol-Q10 and 2 mg/mL E-ASH; *n* = 7). The wild-type mice also received normal drinking water (*n* = 7). Dosage was controlled by initial logging of the amount of water each mouse drank per day (~4 mL/day) The average was used to create the concentration of solution for daily intake. After solution was administered, daily logs were kept ensuring no mouse was intaking significantly more solution than other cohorts. The animals were each housed individually, at 18–23 °C and 40–70% humidity. The animals were placed on a feeding schedule of 8 g of Purina LabDiet Rodent 5001 Chow (North American Lab Supply, Forth Worth, TX, USA) with free access to the drinking solutions that corresponded to the condition they were randomly assigned to. The mice received their food allocation only after behavioural testing was complete, and remaining food was removed several hours prior to the beginning of experiments, to allow mice to be relatively hungry and motivated for reinforcement within the maze. Weights of the animals were recorded daily. To ensure the mice were awake for behavioural assessments during the day, they were under a 12-h light–dark cycle. Behavioural assessments were completed in batches of 4–5 mice per day, two sessions per day. The same procedure was replicated over two phases with two test days per phase.

### 2.3. Behavioural Testing

Higher order cognitive functioning, such as attention and decisiveness, and spatial and emotional memory, are key responsibilities of the prefrontal cortex and hippocampal region of the brain, respectively [58,59]. Neuronal and synaptic degeneration in these brain regions can be detected through memory experiments involving higher-order cognitive processes. For example, spatial and non-spatial working memory in rodents has been evaluated by previous preclinical research through habituation-dishabituation effects [22,60,61]. This is supported by the Sometimes-Opponent Process and the Comparator models, which explain that habituation occurs as organisms learn to expect when, where, and in which context a stimulus will occur over repeated exposures, and that habituation is critical in differentiating between a novel stimulus either received with or in replacement of a habituated stimulus to evoke dishabituation [62,63,64,65]. The current study modified the procedure by Muthukumaran et al. (2017) [22] by implementing the NLR and NOR test within an X-maze. Behavioural assessments were performed at 7 months of age for Phase 1 and 9 months of age for Phase 2 to monitor changes throughout the progression of AD. To evaluate these changes, memory was tested using an X-maze. In the maze, one arm has a distinct background pattern as a distal cue for the mice to follow their position in the maze. The end of each arm contains double-barrelled food cups. The top cup holds a Cheerio morsel which is accessible to the mice, while the bottom cup holds an additional, inaccessible, Cheerio morsel to control for scent cues. The treat foraging procedure motivates mice to maintain consistent locomotor activity throughout repeated exposure to the maze. Exploration was recorded by an infrared camera and analysed using Noldus EthoVision XT 14.0 video tracking software.

The methodology was structured into two phases, each containing two experimental days. Each experimental day consists of 2 sessions. During the first session, mice were exposed to the unchanged maze for 4 consecutive trials, followed by a 2-h intersession interval while other mice completed their trials. The second session on the same day resumed with the same order of mice, continuing with 2 additional unchanged trials. Then, during the third trial of the second session, the NOR test was conducted, in which one habituated object was replaced with a new object to evaluate non-spatial memory. In the second session’s fourth trial, the NLR test was introduced, where one unchanged object was moved to the previously unoccupied arm to assess spatial memory. The next day, the entire procedure was repeated, completing Phase 1. Two months later, the same two-day experiment was conducted again, marking the completion of Phase 2. This experiment was repeated for each mouse, with each trial lasting 3 min. The experimental schedule is detailed in Figure 1.

The purpose of Session 1 is to allow habituation. These 4 trials were replicated without change to the maze environment and referred to as “No Change” (NC) trials. Here, 3 objects, each measuring 3 in × 3 in, were placed in three arms of the maze, leaving one arm empty. A trial began when the animal was placed in the center of the maze and the recording started. Mice were allowed to explore freely for 3 min before the recording ended and the mouse was removed. Inter-trial intervals of 2 min allowed researchers to reset the maze by sanitizing with Prevail cleaning solution to remove odor cues left by the mouse. The animal was then placed back in the center of the maze for the beginning of the next trial. Session 2 Trials 1-2 remained unchanged from the Session 1 NC trials, with Trial 1 serving as a re-habituation trial, and Trial 2 as the final baseline habituation measurement. Session 2, Trial 3, referred to as the “Novel Object” (NO) trial, where the NOR test is employed. Session 2, Trial 4, referred to as the “Novel Location” (NL) trial, with the NLR test. Figure 2 outlines a visual representation of the experimental set-up. To determine changes in exploration rate in the object areas both before manipulation and after, the rate of exploration was considered as a percentage of total time within the maze (3 min). To calculate the exploration rate within each object area, we considered the amount of time the mouse had their nose-point and/or center body point within the 3 in × 3 in object area; however, to account for the time the mice may have been sitting on top of the object, rather than exploring it, we subtracted the time their center body-point was within a 1 in central area within each object area. Next, to assess differences in performance between the two phases, a percent difference in exploration in the manipulated object area versus the same area on the last habituation trial was considered using the following equation:$$Percent\;Difference=\frac{\left(A-B\right)}{B}\times 100\%$$
where *A* is equal to the rate of exploration rate in the manipulated object area on the test trial, and *B* is the rate of exploration in the unmanipulated object area on the last habituation trial. Mean values of each measure were taken per group and phase.

If the hippocampus and related spatial/non-spatial memory functions are maintained, the mouse is expected to recognize these changes in their environment, and thus show an increase in exploration within the novel area compared to the subject’s exploration of the same area during the last habituation trial. In other words, we expect that when the habituated object is moved to a new location within the maze (NL) or replaced with a new object (NO), the wild-type mice and transgenic treated mice (Ubisol-Q10, E-ASH, and Tonic) would show an increase in their exploration within this area. Moreover, the untreated transgenic mice are predicted to show little to no increase in their exploration of the manipulated object area in the NLR test. It is also expected that the untreated mice will show a decline in their performance on the NLR test over time as the disease continues to degenerate their spatial memory, whereas the treated transgenic and wild-type group’s performances are expected to improve or remain the same over time. At the timepoints at which the subjects are evaluated (7 months and 9 months of age for Phase 1 and Phase 2, respectively), it is predicted that Alzheimer’s-type degeneration has not yet affected non-spatial memory. Therefore, we do not expect significant differences amongst the groups for either Phase 1 or Phase 2.

It should be noted that some video file corruption was experienced which led to the loss of around 10% of data points. To account for this, an average from each trial over all trials of the same type were taken per subject to account for the missing points on an individual performance level. This could have led to some level of error; however, the number of corrupted files per subject was minimal, and correction of missed points by subject-specific assessments rather than population-wise aimed to allow a more normalized distribution of individual progress over time.

### 2.4. Statistical Analyses

Statistical analyses were conducted using IBM SPSS Statistics (Version 29), with the statistical significance threshold set at *p* < 0.05. To assess within-subject changes in exploration behavior, paired samples *t*-tests were conducted comparing exploration rates during the final habituation trial (Session 2, Trial 2) to the manipulation trials: the NO test (Session 2, Trial 3) and the NL test (Session 2, Trial 4). These comparisons were conducted separately for each experimental group and for each phase. All paired *t*-tests were two-tailed. Effect sizes for the *t*-tests were calculated using Cohen’s *d* and corrected using Hedges’ *g* to adjust for small sample bias. To evaluate changes in within-subject exploration rates between phases (Phase 1 vs. Phase 2), a one-way ANOVA was conducted for each group. These analyses tested whether the percentage difference in exploration rate from NC to the manipulation trials differed across experimental timepoints. All ANOVA tests were two-tailed.

### 2.5. Tissue Preparation for Immunofluorescence

The mice were euthanized at 18 months of age while under anesthetization using 3% isoflurane at an oxygen flow rate of 2 L/min. Once animals lacked a withdrawal reflex indicating stage 3 anesthesia was reached and absence of pain, they were perfused with ice cold 1X PBS containing 28 μg/mL heparin (Sigma-Aldrich, Oakville, ON, Canada, Cat. No. H3393). This was followed by tissue fixation with ice-cold 4% formaldehyde made in 1X PBS. The brain tissues were dissected and stored in 4% formaldehyde solution at 4 °C. For cryoprotection, the brains were submerged, prior to sectioning, in 30% sucrose (*w*/*v* in PBS) until the tissues sank. Once the brain stem and cerebellum were removed, the tissues were submerged in Shandon™ M-1 embedding matrix (Thermo Scientific, Mississauga, ON, Canada, Cat. No. 1310 TS) before being placed in a vacuum chamber for 10 min. Finally, the brain tissues were cryosectioned coronally at 30 μm and placed onto glass microscope slides.

### 2.6. Immunofluorescence Procedure and Antibodies

Tissue sections were each washed for 5 min in 1X Tris-Buffered Saline (TBS). To enhance the antigen exposure, the tissue sections were then incubated with HistoReveal (Abcam Inc., Cat. No. ab103720) for 5 min. Following this, they were washed for a second time using the 1X TBS. To prevent non-specific binding, the tissue sections were blocked for 1 h with DAKO serum-free protein block (Agilent Technologies Canada Inc., Mississauga, ON, Canada, Cat. No. X0909). Primary antibodies were prepared in DAKO antibody diluent (Agilent Technologies Canada Inc., Mississauga, ON, Canada, Cat. No. S0809) and 1 μL of 10% Tween per 100 μL of total antibody solution. Several primary antibodies were used at the indicated dilution ratio to assess the levels of various proteins: Aβ antibody (mouse IgG, 1:500; Novus Biologicals, Toronto, ON, Canada, Cat. No. NBP2-13075), ionized calcium-binding adapter molecule 1 (Iba-1) (rabbit IgG, 1:300; Novus Biologicals, Toronto, ON, Canada, Cat. No. NB100-1028), beclin-1 (mouse IgG, 1:500; Santa Cruz Biotechnology, Dallas, TX, USA, Cat. No. sc-48342), LC3B (rabbit IgG, 1:500, Abcam Inc., Cambridge, UK, Cat No. ab192890), 4-hydroxynonenal (4-HNE) (rabbit IgG, 1:500; Abcam Inc., Cat. No. ab46545), glial fibrillary acidic protein (GFAP) (rabbit IgG, 1:500; Novus Biologicals, Toronto, ON, Canada, Cat. No. NB300-141), neuronal nuclei (NeuN) antibody (mouse IgG, 1:600; EMD Millipore, Mississauga, ON, Canada, Cat No. MAB 377X), brain-derived neurotrophic factor (BDNF) (rabbit IgG, 1:200; Santa Cruz Biotechnology, Cat. No. sc-20981), and Synaptosomal-associated protein 25 kDa (SNAP25) antibody (rabbit IgG, 1:500; Abcam Inc., Cat. No. ab109105). Following their preparation, the tissue sections were incubated in primary antibody solution over night at 4 °C.

The next day, tissues sections were each washed twice for 5 min with 1X TBS. Secondary antibody solutions were prepared following the same protocol as the primary antibodies. The following secondary antibodies were used at a 1:500 dilution, depending on the primary antibody that had been used: Alexa FluorTM 568 goat anti-rabbit IgG (Thermo Scientific Canada, Mississauga, ON, Canada, Cat. No. A-11011) and FluorTM 488 donkey anti-mouse IgG (Thermo Scientific Canada, Mississauga, ON, Canada, Cat. No. A-21202). At room temperature, the tissue sections were incubated in the secondary antibody solution for 2 h. Following incubation, sections were washed twice for 5 min with 1X TBS for a second time. The microscope slides were then coverslipped with VECTASHIELD^®^ VibranceTM Antifade Mounting Medium with DAPI (MJS BioLynx Inc., Brockville, ON, Canada, Cat. No. VECTH180010) and were allowed to cure for a minimum of 1 h.

Imagining under epifluorescence with the Leica DMI6000 B Inverted Microscope (Leica Microsystems, Concord, ON, Canada) was conducted, where three fields using different sections for each animal were captured. Three fields were imaged per group and the average corrected total fluorescence (CTF) was quantified using ImageJ 1.54 software and average CTF’s were graphed as a % of control group using Microsoft Excel. It should be noted that quantification of fluorescent intensity relative to the control is strictly meant to visualise the general trend of qualitative results. Therefore, no statistical analysis was conducted on this data.

## 3. Results

### 3.1. Effects of Treatment with Ubisol-Q10 and E-ASH on Spatial Memory

To assess spatial memory, the NLR test was utilized. A series of paired samples *t*-tests were conducted to compare exploration rates of the manipulated object area (NL) with the previous habituation trial (NC) within each treatment group across Phase 1 and Phase 2. Results show (see Figure 3a) that exploration of the manipulated object location generally increased across all groups and phases. As predicted, significant increases in exploration of the manipulated object area compared to the habituated area in Phase 1 was seen for E-ASH (*t*(15) = −3.55, *p* = 0.003, *d* = 0.89), Tonic (*t*(13) = −2.73, *p* = 0.017, *d* = 0.73), and wild-type (*t*(13) = −2.51, *p* = 0.026, *d* = 0.67) groups. Untreated mice also had a significant increase in exploration in Phase 1, *t*(11) = −3.17, *p* = 0.009, *d* = 0.92; however, the significance was not carried through to Phase 2, *t*(11) = −2.00, *p* = 0.071, *d* = 0.58. Significant increases in exploration of the manipulated object area in Phase 2 were seen in the Ubisol-Q10 (*t*(11) = −2.26, *p* = 0.045, *d* = 0.65), Tonic (*t*(13) = −2.25, *p* = 0.042, *d* = 0.60), and wild-type (*t*(13) = −2.30, *p* = 0.039, *d* = 0.61) groups. Next, comparing between phases as a percent difference in exploration rate (Figure 3b), a one-way ANOVA was conducted. There were no significant differences in performance across all groups (E-ASH, *F*(1, 30) = 1.06, *p* = 0.311, η^2^ = 0.03; Ubisol-Q10, *F*(1, 22) = 0.43, *p* = 0.521, η^2^ = 0.02; Tonic, *F*(1, 26) = 0.11, *p* = 0.745, η^2^ = 0.004; untreated, *F*(1, 22) = 1.29, *p* = 0.269, η^2^ = 0.06; wild-type, *F*(1, 26) = 0.51, *p* = 0.481, η^2^ = 0.02). Looking at the general trend, untreated mice stand alone in showing a decline in their performance on the NLR test. The Tonic group shows an increase, and the other groups show little difference between phases. These results imply that the treatments, both independently and combined, allowed transgenic AD mice to perform spatial memory tasks at par with wild-type mice, while untreated mice predictably declined. In particular, the Tonic-treated mice performed above independently treated mice.

### 3.2. Effects of Treatment with Ubisol-Q10 and E-ASH on Non-Spatial Memory

The NOR test was used to assess non-spatial memory by replacing a well-habituated object with a new object. Paired samples *t*-tests were conducted to determine whether exploration increased from the final habituation trial to the NO test within each treatment group across Phase 1 and Phase 2. Results showed (see Figure 4a) that exploration of the manipulated object area increased across all groups and phases. Phase 1 mean values of rate of exploration significantly increased for E-ASH (*t*(15) = −3.61, *p* = 0.003, *d* = 0.90), Ubisol-Q10 (*t*(11) = −5.11, *p* < 0.001, *d* = 1.48), and wild-type (*t*(13) = −3.17, *p* = 0.007, *d* = 0.85, and Phase 2) groups. Untreated mice were not seen to have a significant increase in exploration in the Phase 1 NOR test (*t*(11) = −1.50, *p* = 0.162, *d* = 0.43), nor in Phase 2 (*t*(11) = −1.82, *p* = 0.097, *d* = 0.52). Mean values for rate of exploration of the NO area in Phase 2 were seen to significantly increase for E-ASH (*t*(15) = −2.22, *p* = 0.043, *d* = 0.55), Ubisol-Q10 (*t*(11) = −3.32, *p* = 0.007, *d* = 0.96), Tonic (*t*(13) = −3.06, *p* = 0.009, *d* = 0.82) and wild-type (t(13) = −2.51, *p* = 0.026, *d* = 0.67) groups. Furthermore, comparing between phases (Figure 4b), a one-way ANOVA revealed there were no significant differences in performance via percent difference in exploration rate across all groups (E-ASH, *F*(1, 30) = 1.77, *p* = 0.193, η^2^ = 0.06; Ubisol-Q10, *F*(1, 22) = 0.07, *p* = 0.799, η^2^ = 0.003; Tonic, *F*(1, 26) = 0.16, *p* = 0.696, η^2^ = 0.006; untreated, *F*(1, 22) = 1.56, *p* = 0.224, η^2^ = 0.07; wild-type, *F*(1, 26) = 0.15, *p* = 0.701, η^2^ = 0.006). The trend showed that E-ASH-, and Ubisol-Q10-treated mice decreased in their performance of the NOR test, whereas Tonic, untreated and wild-type mice increased in their performance between phases. The results suggest that Ubisol-Q10 and E-ASH provided a performance on the NOR test comparable to wild-type mice, whereas the untreated AD mice showed no significant difference in exploration in either Phase 1 or Phase 2.

### 3.3. Effect of Treatment with Ubisol-Q10 and E-ASH on Aβ Plaque Load and Staining for Microglia in the Brains of Double Transgenic AD Mice

It has been previously reported that the presence of Aβ plaques leads to the activation of microglia [22,29,30]. As seen in Figure 5, wild-type mice did not show any staining for Aβ plaques due to the lack of the APP gene and showed minimal staining of microglia. The positive control, transgenic mice given regular drinking water, showed the highest quantity and largest sizes of Aβ plaques, as well as the greatest amount of staining for microglia. Transgenic mice given Ubisol-Q10 supplemented drinking water had fewer Aβ plaques, decreased Aβ plaque size, and reduced microglia compared to the positive control. Transgenic mice treated with E-ASH had reduced staining for microglia and reduced Aβ plaque number compared to the positive control, but Aβ plaque size appeared unchanged. Transgenic mice treated with the Tonic solution had a reduction in Aβ plaque size and quantity compared to when either Ubisol-Q10 or E-ASH were used alone, as well as reduced staining for microglia.

### 3.4. Activation of Autophagy via Ubisol-Q10 and E-ASH Treatment

We have previously shown that the autophagy regulator beclin-1 was upregulated in fibroblasts from AD patients and the brains of double transgenic mice treated with Ubisol-Q10. We have also shown that beclin-1 is upregulated in Ubisol-Q10- and E-ASH-treated animals using a Parkinson’s disease rat model, a neurodegenerative disorder with similar mechanisms to AD. In Figure 6, here we probed for beclin-1 and autophagosome marker LC3B, which is also shown to be modulated in response to Ubisol-Q10. Untreated, transgenic mice showed reduced beclin-1 levels compared to wild-type mice. Transgenic mice given Ubisol-Q10 showed elevated beclin-1 expression, increased relative to wild-type mice. E-ASH-treated transgenic mice appeared similar to untreated mice in expressing both autophagy-related proteins. Transgenic mice fed water supplemented with the Tonic solution showed elevated expression of beclin-1, similar to wild-type mice and transgenic mice fed Ubisol-Q10.

### 3.5. Reduction in Oxidative Stress with Ubisol-Q10 and E-ASH Treatment

It is well known that levels of reactive oxygen species and oxidative stress are elevated in AD. Previously we have seen that 4-HNE, a lipid peroxidation product and marker of oxidative stress, is elevated in AD fibroblasts and double transgenic mice. In this experiment, we investigated the effect of treatment with Ubisol-Q10 and E-ASH on 4-HNE presence in the brains of double transgenic mice, and results are shown in Figure 7. Wild-type showed minimal expression for 4-HNE. Untreated transgenic mice showed elevated levels of 4-HNE compared to wild-type mice, while treated transgenic mice showed reduced presence of 4-HNE, with the greatest reduction in the Ubisol-Q10 and Tonic groups.

### 3.6. Effect of Treatment with Ubisol-Q10 and E-ASH on Astrocyte Activation and NeuN in the Brains of Double Transgenic AD Mice

Previously, the CA1 layer and astrocyte activation was shown to be reduced in double transgenic mice. Here we wanted to see if another region of the hippocampus, the CA3, was affected in double transgenic mice. We also wanted to investigate the status of astrocytes in response to Ubisol-Q10 and E-ASH treatment. As seen in Figure 8, there was no observable difference in the groups’ thickness for NeuN in the CA3 region, however an increase in fluorescence intensity was seen in the Tonic condition. Wild-type and transgenic mice given Ubisol-Q10 showed higher levels of staining for astrocytes compared to transgenic untreated. Transgenic mice given E-ASH or the Tonic solution had a greater expression of staining for GFAP, a marker for astrocytes that has been shown to be upregulated in their active form, compared to mice given the other treatments.

### 3.7. Effect of Ubisol-Q10 and E-ASH on Levels of BDNF

Following an investigation of astrocyte status, we probed for BDNF, a neurotrophic factor released by active astrocytes, to determine its association. As seen in Figure 9, untreated transgenic mice showed reduced expression of BDNF compared to wild-type mice, similar to GFAP in Figure 8. Transgenic mice treated with Ubisol-Q10, E-ASH, or the Tonic solution showed elevated expression of BDNF compared to untreated mice.

### 3.8. Effect of Ubisol-Q10 and E-ASH on Synaptic Health Marker SNAP-25

The density of synaptic connections between neurons in the hippocampus is a biomarker of AD progression. The final analysis assessed the preservation of synaptic vesicle formation using the marker SNAP-25, a SNARE complex protein important in calcium-gated neurotransmitter release. As seen in Figure 10, synaptic vesicle formation was preserved across all transgenic mice treated with Ubisol-Q10, E-ASH, or Tonic compared to the untreated transgenic group. Tonic-treated mice showed enhanced expression of SNAP-25 compared to untreated mice. Additionally, synaptic density was greatly improved in treated groups compared to the wild-type control.

## 4. Discussion

Due to the complexity of AD, we suggest that a multifaceted treatment approach is preferrable. Following treatment with Ubisol-Q10 and E-ASH, we saw reductions in the size and quantity of Aβ plaques, active microglia, and oxidative stress; elevated expression of autophagy markers, astrocyte activation, BDNF, and NeuN fluorescent intensity; and a preservation of synaptic vesicle formation and improvements in the synaptic density. Pre-mortem, spatial and non-spatial memory tests suggest Ubisol-Q10 and E-ASH treatment, either independently or in combination, hold potential to preserve cognitive functioning long-term. Taken together, this suggests that Ubisol-Q10 and E-ASH treatment in a transgenic mouse model addresses many of the biochemical mechanisms and cognitive deficits characteristic of AD.

Initial phytochemical analysis revealed the composition of our E-ASH [56]. It is evident that the concentration of withanolides greatly outweighs that of flavonoids. Withaferin A is particularly dominant in this solution, which is known for anti-stress activity and immunomodulatory effects [51]. Withaferin A from ASH root has been shown to significantly reverse cognitive declines caused by ibotenic acid in AD models [51]. The active property of ASH works to mitigate inflammation through inhibition of IκB phosphorylation and NFκB-mediated transcription [54]. Since the extract comprises 12.4% withanolides, this is assumed to be the dominant player in E-ASH neuroprotective mechanisms. Withanolide A is also of higher concentration in this extract, which has been shown to promote axonal regeneration and synaptic restoration, improve memory, and promote neurite outgrowth in various neurodegenerative models [66]. While the flavonoid content is not as potent, it cannot be disregarded for having supporting properties which aid in neuroprotection. It is important to note that we do not aim to isolate each active compound within this extract and determine their exact role in the treatments efficacy, because the positive results come from the complex, unique, and inimitable interactions between all components of this natural product. These qualities have been supported throughout the results of the current study through E-ASH-treated transgenic mice showing obvious reduction in activated microglia, improved astrocytic activity, preserved neuronal and synaptic integrity, decreased oxidative stress and improved memory processes. We will expand on the specifics of E-ASH qualities, and how they interact with Ubisol-Q10 to mitigate the effects of AD neurodegeneration in the following discussion.

One of the key pathological features of AD is the development and accumulation of Aβ plaques [11,12,13] that can lead to neuronal death [67,68]. Presently, we have demonstrated the neuroprotective efficacy of using a combined treatment of Ubisol-Q10 with E-ASH in a double transgenic mouse model of AD, where Tonic treatment resulted in enhanced Aβ plaque clearance. Treatment with Ubisol-Q10 and E-ASH independently also reduced the quantity of Aβ plaques compared to the untreated group. This is consistent with previous work evaluating the effect of Ubisol-Q10 treatment on Aβ plaques in the brains of transgenic mice [22]. We suggest this Aβ plaque clearance may be facilitated by the treatment-mediated resumption of autophagy activation. The present study notes a reduction in the expression of autophagy markers of beclin-1 and LC3B in the untreated double transgenic mice in comparison to the wild-type control. This is consistent with findings that beclin-1 is reduced in patients with AD and that reduced expression of the protein has been associated with autophagy in a mouse model [69]. Further, we note that treatment with Ubisol-Q10, both independently and in Tonic solution, results in an upregulation of beclin-1 and LC3B, which is also consistent with previous findings [19,24]. Taken together, these results are consistent with a current hypothesis suggesting autophagy inhibition is a biochemical feature of the disease [24,26,27,70] and suggest that Ubisol-Q10 might promote the resumption of this process, which in turn helps to drive the clearance of Aβ plaques. Interestingly, E-ASH independently was not found to have the same effect on autophagy, thus it may be working to reduce Aβ plaques via a different protein clearance pathway. Although the involvement of an additional protein clearance pathway by E-ASH still needs to be evaluated, were this the case, it would explain the enhanced Aβ clearance observed with Tonic treatment. With the results showing increased astrocytic activation (GFAP and BDNF), reduced macroautophagy markers (beclin-1 and LC3B), and slightly decreased Iba-1 expression in E-ASH-treated mice, this suggests macroautophagy or microglial phagocytosis may not be the primary pathways for Aβ clearance. Instead, astrocyte-mediated mechanisms, such as astrocytic uptake and degradation, or Aβ-degrading protease secretion, may be responsible for the reduced Aβ plaque load [53].

Additionally, it has been shown that Aβ plaques activate surrounding microglia [22,29,30], leading to the release of pro-inflammatory cytokines [33] that increases cellular stress. Treatment with Ubisol-Q10 or E-ASH reduced active microglia as indicated by a reduction in staining for Iba-1 compared to untreated transgenic mice. This is consistent with previous studies that have found Ashwagandha leaf extract inhibits microgliosis, suppressing the subsequent production of pro-inflammatory cytokines, such as interleukin-1 beta (IL-1β) and interleukin-6 (IL-6) [54]. The Tonic condition saw the greatest reduction in active microglia. The reduction in the size and number of Aβ plaques following Ubisol-Q10 treatment may result in fewer microglia activated by the toxic protein aggregates. This notion is consistent with findings that the production of pro-inflammatory cytokines by microglia is dependent on Aβ peptide concentration in-vitro [71]. This decrease in active microglia may also be related to the increased beclin-1 expression observed in mice treated with Ubisol-Q10, as Houtman et al. demonstrate that the production of pro-inflammatory cytokines, IL-1β and interleukin-18 (IL-18), is enhanced with reduced beclin-1 expression [72]. When this reduction in beclin-1 is mitigated, as we noted with Ubisol-Q10 treatment, there may be a decrease in production of pro-inflammatory cytokines indicating fewer active microglia.

In contrast to pro-inflammatory microglia, astrocytes provide support to neurons, including maintaining homeostasis, controlling the blood–brain barrier, and supporting synapses [73,74,75]. While we observed astrocyte activation in the untreated transgenic group, this is not unexpected, as previous studies have reported increased reactive astrocytes around Aβ plaques [76]. We saw elevations in the expression of astrocyte activation in the groups receiving treatment, in comparison to the untreated transgenic mice, as indicated by increased staining for GFAP and a greater number of astrocyte processes extending from the cell body. Previous studies have also shown the efficacy of ASH in enhancing astrocyte activation [54], and we suggest that the treatment is acting in the same manner here. The role astrocytes play in the pathology of AD is complicated, with their activation having the potential to be both neuroprotective and neurotoxic [76,77]. Neurotoxic active astrocytes are thought to be induced by the release of pro-inflammatory cytokines from active microglia [78,79], as blocking astrocyte activation by microglia has been shown to be neuroprotective in transgenic mouse models [80]. Given we saw reductions in the activation of microglia following treatment, we suggest the astrocyte activation noted is neuroprotective. The observed reduction in Aβ plaque load also points to the neuroprotective contribution of astrocytes, which are known to be involved in the clearance of the protein aggregates [76,81]. However, it is important to note that further work evaluating the expression of pro-inflammatory cytokines released by active microglia known to activate neurotoxic astrocytes (e.g., Il-1α, TNFα, and C1q) [79], is needed.

It has also been suggested that, with AD, long-term exposure of astrocytes to Aβ results in the impairment of their other supportive functions in favour of plaque clearance [81]. With treatment, plaque clearance driven by astrocyte-independent mechanisms may aid in amplifying their neuroprotective effect by freeing them to carry out other roles. For example, active astrocytes promote neuronal survival and functioning through the release of neurotrophic factors [73,74,77], including BDNF, which has been shown to be neuroprotective and may be involved in reducing Aβ plaque size [82,83,84,85,86]. We observed enhanced expression of BDNF in all treatment groups, in comparison to untreated transgenic mice. Given we also saw an increase in active astrocytes in the treated mice, it is unsurprising that expression for BDNF followed the same pattern, suggesting the increase in active astrocytes may have driven an increase in the release of BDNF. Also, all treatment groups with upregulated levels of GFAP (E-ASH and Tonic groups) displayed intense upregulation of NeuN and BDNF, thus indicating that the treatments are activating a neuroprotective astrocytic pathway. Ubisol-Q10-treated mice also showed upregulated BDNF and NeuN; however, GFAP staining showed reduced levels of astrocytic activity, and therefore may be aiding in neuroprotection through a different pathway, such as improved autophagy, which was noted earlier as being an absent Aβ clearance pathway for E-ASH. Therefore, the Ubisol-Q10 and E-ASH may be enhancing neuroprotection through different pathways. However, in future studies, immunofluorescent staining of BDNF and GFAP together should be conducted to see how their expression might correlate.

AD is also characterized by the generation of ROS and elevations in oxidative stress [19,21,22,23,24,25], which are known to trigger apoptosis at high concentrations [87]. Previously we have seen that 4-HNE, a marker for oxidative stress, is elevated in AD fibroblasts and double transgenic mice [22,24]. Presently, we evaluated the efficacy of Ubisol-Q10 and E-ASH on relieving oxidative stress, as evaluated by 4-HNE. Treatment with Ubisol-Q10 showed the greatest reduction in oxidative stress, in comparison to untreated transgenic mice; however, reduced 4-HNE expression was also noted in E-ASH and Tonic conditions. This is consistent with previous reports of Ubisol-Q10 and E-ASH expressing antioxidant properties [88,89]. Interestingly, Aβ is involved in a positive feedback loop in which oxidative stress leads to increases in Aβ [87,90], which also drives increased oxidative stress [21,25,87,90]. While we observed decreases in both Aβ and oxidative stress when transgenic mice were treated with Ubisol-Q10 and E-ASH, it is unclear where in this cycle treatment was able to intervene. Nonetheless, it is possible the breaking of this feedback loop amplified the efficacy of the treatments.

One of the key features of AD pathology is progressive neuron loss in the hippocampus and cortex [5], with previous studies having reported substantial decreases in neuron number and volume in AD patients [91]. Some of these decreases were seen in the CA1 [91,92], and CA3 region of the hippocampus [92]. Since we have previously reported reductions in the number of neurons in the CA1 pyramidal region of the brains of transgenic mice in comparison to those treated with Ubisol-Q10 [22], here we assessed whether CA3 neurons are reduced in double transgenic AD mice. With immunofluorescent staining, we did not see qualitative differences in the thickness of the CA3 region for any of the groups. However, greater fluorescence intensity was detected in the Tonic condition compared to untreated mice. More research is required to assess the effects of Aβ on the CA3 pyramidal layer, as other transgenic mice show neurodegeneration in this region [93].

Additionally, in patients with AD, it is believed that synaptic health is negatively affected, where the decreases in synaptic connectivity seen throughout disease progression correlate with cognitive decline [1,2,3,4]. Reduced SNAP-25 mRNA expression has been reported in both AD mouse models [94] and AD patient brain tissue [95]. Presently, there was a slight reduction in the protein’s expression in the untreated transgenic mice, in comparison to the wild-type control; however, this is consistent with a previous report, which saw approximately 10% reductions in the brains of AD patients [96]. Importantly, reductions in SNAP-25 expression have been shown to have negative impacts on synaptic health, leading to functional deficits and reduced density of dendritic spines [97,98]. When taken into consideration alongside previous research implicating Aβ in the loss of dendritic spines [99,100], these mechanisms may provide some insight into the concerns of synaptic health in AD patients. Since treatment with the Tonic solution increased both SNAP-25 expression and Aβ clearance, it is suggested that it will help to correct for dendritic spine loss, which in turn may help to alleviate some of the cognitive deficits correlated with the loss of synaptic connections [2,3]. Thus, we have shown the efficacy of treatment, specifically with Ubisol-Q10 and Tonic solution, in preserving synaptic functioning. We observed an upregulation higher than the wild-type group, which is presumed to be positive; however, given the role SNAP-25 plays in neurotransmission, it is suggested that future studies confirm there are no excitotoxic effects of the expression levels seen.

Moreover, a previous study evaluated wild-type mice against Ubisol-Q10-treated transgenic mice and untreated transgenic mice in a Y-maze to evaluate habituation over repeated exposure to unchanged objects within the maze, and dishabituation through NOR and NLR tests [22]. Therefore, the current study aimed to expand on this to explore the effects of Ubisol-Q10 and E-ASH both independently and in combination following the same NOR and NLR tests in a modified X-maze.

Confirming our predictions of the NLR test, the Tonic-treated transgenic and wild-type groups maintained significant increases in exploration of the manipulated object in both phases, while untreated transgenic mice failed to carry significant exploratory performance in the NLR task moving into Phase 2. Ubisol-Q10-treated transgenic mice also displayed a significant increase in exploration in Phase 2 but not Phase 1. This compliments the results found by Muthukumaran et al. (2017) [22], where the Ubisol-Q10-treated transgenic (both in combination and independently administered in our current study) and wild-type groups remained comparable in later degenerative stages. Against our expectations, the E-ASH-treated transgenic group did not maintain a significant increase in their exploration in the NLR test in Phase 2. This could be indicative that E-ASH has some absence in later-stage spatial memory protection that Ubisol-Q10 is able to fulfill in the Tonic-treated transgenic mice. However, looking at the percent difference in exploration on the NL trial compared to the last NC trial, we can see a clear trend forming. The treated transgenic groups’ performances on the NLR test remained the same or improved between phases, while the untreated transgenic group had a relatively large decrease in their performance on this task. It is also interesting to note that the NLR test results show a resemblance to results from the oxidative stress marker, 4-HNE, where we saw Ubisol-Q10 and Tonic-treated mice had a greater reduction in oxidative stress than E-ASH-treated animals. Therefore, it may be possible the reduction in oxidative stress via Ubisol-Q10 is aiding in the longitudinal maintenance of non-spatial memory functions through preserved synaptic integrity. The results suggest that Ubisol-Q10 and E-ASH treatment may have time-dependent effects when administered independently; however, they are able to maintain long-term spatial memory preservation comparable to healthy wild-type mice when administered as a combination Tonic treatment.

Furthermore, the NOR test revealed a general increase in rate of exploration of the manipulated object area when compared to the previous baseline habituation trial. As expected, wild-type, E-ASH-treated transgenic, and Ubisol-Q10-treated transgenic mice had significant increases in their exploration of the novel object throughout both Phase 1 and Phase 2, and Tonic-treated mice displayed significant NOR performance in the later phase. In addition, the untreated transgenic mice failed to meet significant NOR exploratory performance throughout both phases. This trend contradicts previous research by Muthukumaran et al. (2017) [22], where Ubisol-Q10-treated transgenic, untreated transgenic, and wild-type mice all maintained significant increased exploration on the NOR test. However, the current results support that the treatments of Ubisol-Q10 and E-ASH, together and independently, protect non-spatial object recognition skills over the long-term. To compare performance on the non-spatial memory task between phases, a percent difference in exploration from NC to NO trials was calculated and considered. Wild-type, Tonic-treated transgenic and untreated transgenic had greater performance on the NOR test in Phase 2 compared to Phase 1, whereas E-ASH- and Ubisol-Q10-treated transgenic mice displayed a decrease. While this trend goes against our hypothesis, the phenomenon could be due to a plateau of performance over repeated testing. In addition, the standard error of the percent difference in the untreated group is large relative to the other groups, which is indicative of large variability between subjects. This could have potentially been exacerbated due to the ~10% of data loss as mentioned previously, which may introduce time-dependent biases to the data when filled in with mean values from the same subject from the same trial-type, but at a slightly different timepoint. Further testing at later timepoints is necessary to assess the accuracy of the trend.

Future directions of the current study aim to include more longitudinal behavioural analyses. By extending into late-stage neurodegeneration (beyond 9 mo) we can dive deeper into these findings and confirm the treatments’ trajectory of effects on this mouse model. In addition, it should be acknowledged that animal models do not always equate to the human experience, and despite our best efforts to create a human-like testing environment, future clinical studies are necessary to confirm the treatments’ efficacy at human equivalent doses (HED). Treatment dosing for mice is higher than that required for humans, due to metabolic differences between species [101]. Therefore, translating this treatment to humans should be calculated using the formula for HED [101]:$$HED\;\frac{mg}{kg}={\left[Animal\;no\;observed\;adverse\;effect\;levels\;(NOAEL)\frac{mg}{kg}\right]\;\times \left[\frac{Weight\;animal\left(kg\right)}{Weight\;human\left(kg\right)}\right]}^{\left(1-0.67\right)}$$

So, in this case, with mice NOAL being 8 mg/kg/day for Ubisol-Q10 and 320 mg/kg/day for E-ASH, human equivalent dose for a 60 kg human should be 0.613 mg/kg/day for Ubisol-Q10 and 24.53 mg/kg/day for E-ASH. However, human clinical trials will need to be conducted to confirm the translation is effective. If effective, the ease of administration of this Tonic solution may be a preferrable treatment method given its efficacy via oral administration in normal drinking water. In addition, the mice given this Tonic showed no adverse side effects or toxicity, even with constant intake over their lifespan. Therefore, the Tonic may provide a safe, non-invasive, and longitudinally safe treatment option for humans vulnerable to AD.

Altogether, pre-mortem and post-mortem analysis of transgenic mice treated with Ubisol-Q10, E-ASH, and a combination Tonic solution implied that AD-like degenerative mechanisms, both biochemical and cognitive, were preserved equivalent to healthy wild-type mice. The efficacy of the treatments became evident when compared against an untreated AD mouse model, where typical degenerative paths were followed. Figure 11 outlines the mechanistic highlights of each treatment to represent their symbiotic relationship in the Tonic treatment. The current multi-disciplinary longitudinal assessment of the novel nutraceutical therapy to potentially halt the progression of AD resulted in cohesive evidence that Ubisol-Q10 and E-ASH are efficient neuro-protecting agents at both the biochemical and cognitive perspectives.

## 5. Conclusions

Presently, we have demonstrated the efficacy of treatment with Ubisol-Q10 and E-ASH in addressing the pathogenesis of AD in double transgenic mouse models. In comparison to independent treatment, the Tonic treatment showed enhanced Aβ plaque clearance; preservation of synaptic vesicle formation, suggesting increases to overall synaptic health; inhibition of microgliosis; and preserved long-term non-spatial and spatial memory performance. Treatment with Ubisol-Q10, either independently or in combination, showed enhanced activation of autophagy and reductions in oxidative stress. Ubisol-Q10-treated mice significantly outperformed their unprotected sub-group on non-spatial memory tests, and late-degenerative-stage spatial memory tests. Treatment with E-ASH, both independently and in the Tonic condition, lead to an increase in active astrocytes, which are suggested to have been neuroprotective, as well as enhanced BDNF expression. E-ASH-treated mice showed a longitudinal improvement in their non-spatial memory functions compared to untreated AD mice. While there was no observable difference between the thickness of the CA3 region of the hippocampus in any of the mice, greater intensity of fluorescence was seen in treated groups; however, more work is required to fully assess the status of neurons in the pyramidal region of the hippocampus. It is possible that Ubisol-Q10 and E-ASH target different pathologies of AD, such as inflammation, oxidative stress, and autophagy, which results in enhanced removal of these Aβ plaques in the brains of these transgenic mice. Further, we suggest that the impact of treatment on one pathology may work to amplify the effect of treatment on others; however, the specific mechanisms by which Ubisol-Q10 and E-ASH work to target the evaluated pathologies of AD need to be explored to fully understand their enhanced efficacy in combination. Future directions of this study will mechanistically validate the autophagy-related role of this treatment and include more longitudinal behavioural analysis to evaluate the trajectory of treatment efficacy. With the reduction in Aβ plaques, inflammation, and oxidative stress, as well as resumption of autophagy, preservation of synaptic health, and protection of cognitive functions by the combination treatment, the overall progression in AD might be halted. This simple, orally administered, and well-tolerated nutraceutical formulation would be an effective treatment option for AD.

## 6. Patents

Ubisol-Q10 is a patented formulation.

## Figures and Tables

**Figure 1 nutrients-17-02701-f001:**
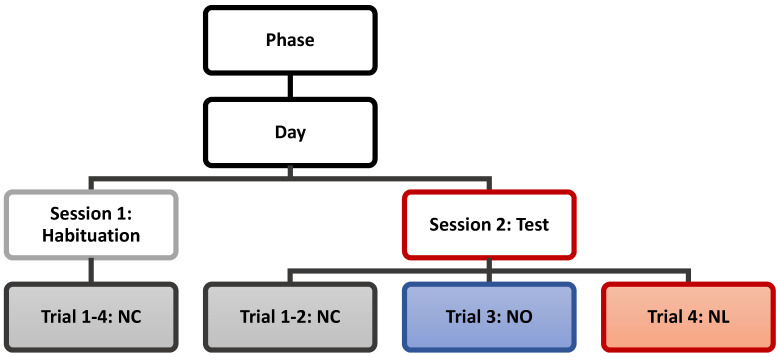
Schematic diagram of experimental procedures per phase; NC = No Change, NO = Novel Object, and NL = Novel Location.

**Figure 2 nutrients-17-02701-f002:**
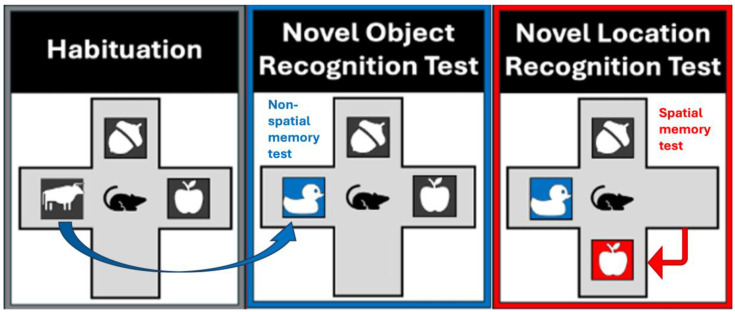
Example of maze setup on habituation and test trials. Arrows represent how the objects change between trials.

**Figure 3 nutrients-17-02701-f003:**
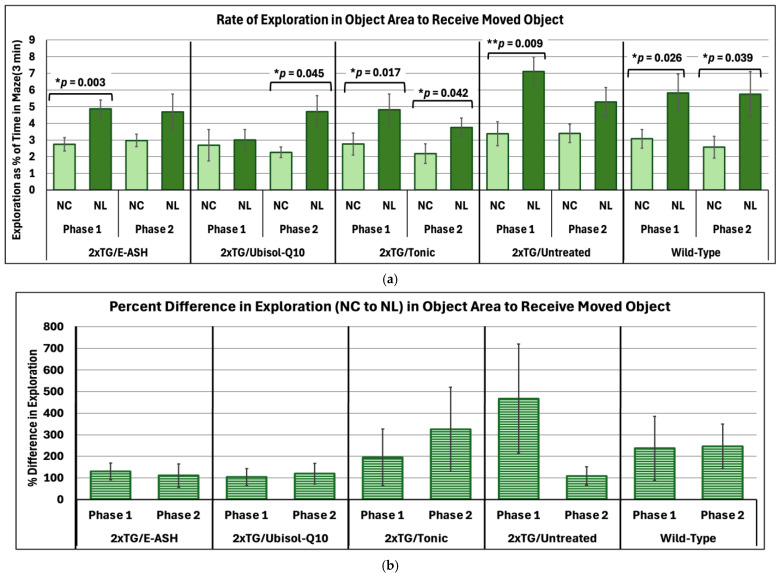
Changes in rate of exploration within object area manipulated during Novel Location test versus No Change trial across phases. (**a**) Mean values of rate of exploration as a percent of total time in the maze (3 min) ± SEM is determined from all subjects within their respective conditions. Rate of exploration is defined by amount of time the subject had their nose-point and/or center body point within the object area while subtracting the time their center body point was within the centermost 1 cm radial area within the object area. NC = No Change, and NL = Novel Location. * *p* < 0.05; ** *p* < 0.01. (**b**) Percent difference in rate of exploration between NC and NL trials was compared within each subject condition between phases. Phase 1 data was collected at 7 months of age, and Phase 2 data was collected at 9 months of age.

**Figure 4 nutrients-17-02701-f004:**
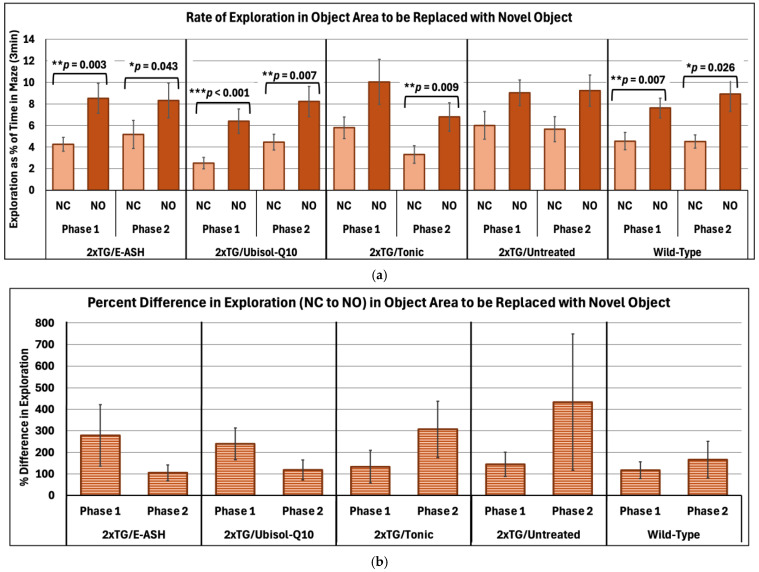
Changes in rate of exploration within object area manipulated during Novel Object test versus No Change trial across phases. (**a**) Mean values of rate of exploration as a percent of total time in the maze (3 min) ± SEM is determined from all subjects within their respective conditions. Rate of exploration is defined by amount of time the subject had their nose-point and/or center body point within the object area while subtracting the time their center body point was within the centermost 1 cm radial area. NC = No Change and NO = Novel Object. * *p* < 0.05; ** *p* < 0.01, *** *p* < 0.001. (**b**) Percent difference in rate of exploration between NC and NO trials was compared within each subject condition between phases. Phase 1 data was collected at 7 months of age, and Phase 2 data was collected at 9 months of age.

**Figure 5 nutrients-17-02701-f005:**
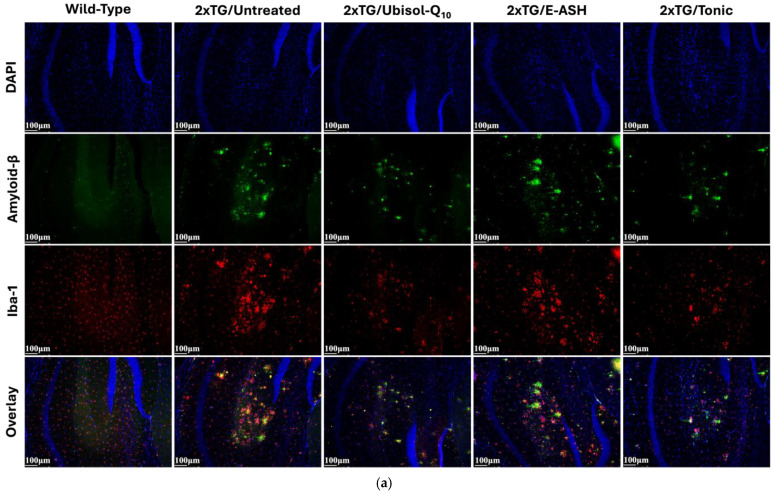
Effect of Ubisol-Q10 and E-ASH on Aβ plaque deposition and Iba-1 expression. (**a**) Immunofluorescent staining in hippocampus probing for Aβ plaques and ionized calcium-binding adapter molecule 1 (Iba-1). (**b**) Representation of fluorescence (corrected total fluorescence; CTF) of Aβ represented as a % of 2 × TG/Untreated with mean values relative to untreated controls are added in the figure legend, and (**c**) Iba-1 represented as a % of wild-type with mean values relative to wild-type controls are added in the figure legend. A total of 3 fields were imaged and mean values ± SEM are graphed. Nuclei were counterstained with DAPI. Micrographs were taken at 200× magnification. Scale bar = 100 microns.

**Figure 6 nutrients-17-02701-f006:**
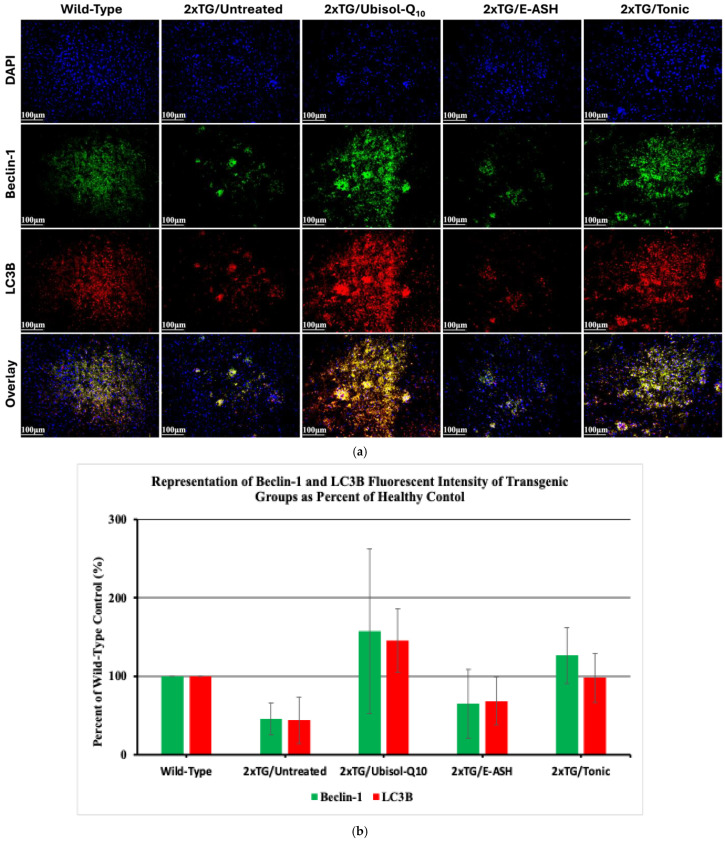
Effect of Ubisol-Q10 and E-ASH on autophagy activity. (**a**) Immunofluorescent staining in hippocampus probing for autophagy regulator beclin-1 and essential autophagy related protein LC3B and (**b**) representation of qualitative fluorescence (corrected total fluorescence; CTF) of beclin-1 and LC3B represented as a % of wild-type. A total of 3 fields were imaged and mean values ± SEM are graphed. Nuclei were counterstained with DAPI. Micrographs were taken at 200× magnification. Scale bar = 100 microns.

**Figure 7 nutrients-17-02701-f007:**
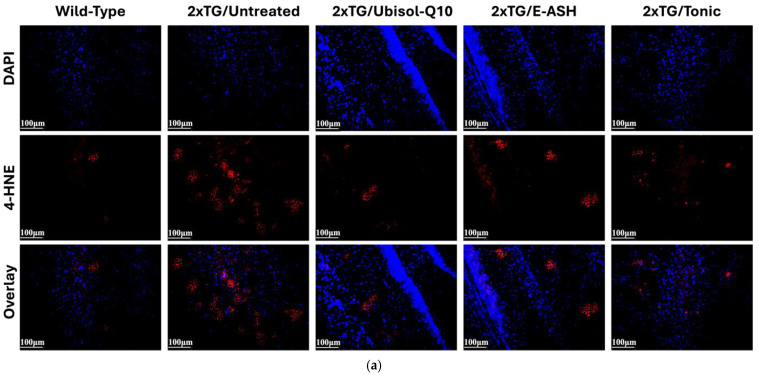
Effect of Ubisol-Q10 and E-ASH on lipid peroxidation. (**a**) Immunofluorescent staining in hippocampus probing for 4-hydroxynonenal (4-HNE), a lipid peroxidation by-product and marker of oxidative stress and (**b**) representation of fluorescence (corrected total fluorescence; CTF) of 4-HNE represented as a % of wild-type. A total of 3 fields were imaged and mean values ± SEM are graphed. Nuclei were counterstained with DAPI. Micrographs were taken at 200× magnification. Scale bar = 100 microns.

**Figure 8 nutrients-17-02701-f008:**
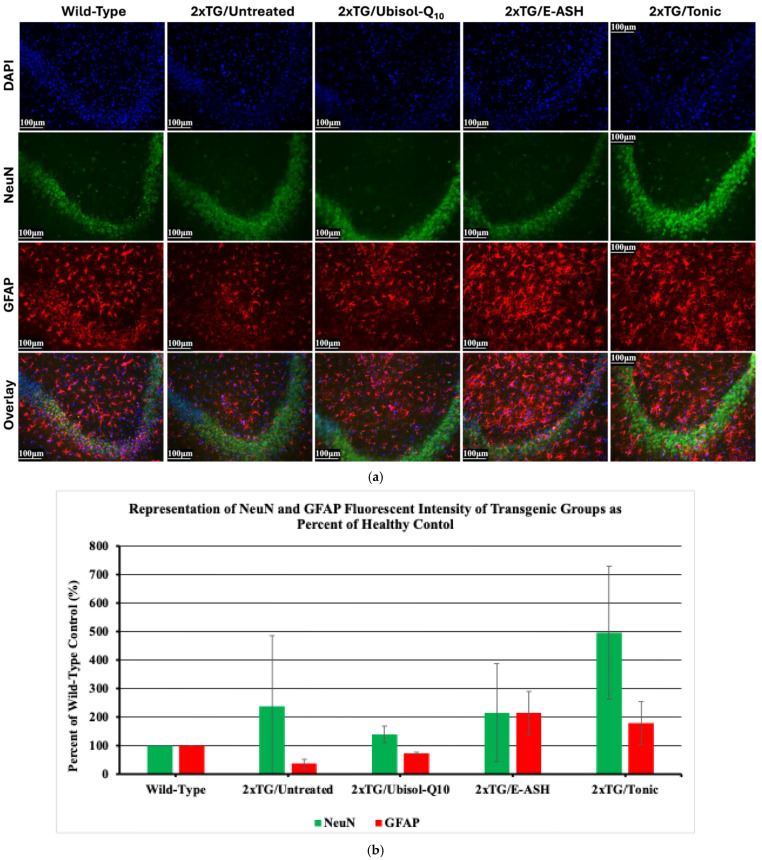
Effect of Ubisol-Q10 and E-ASH on astrocyte activation and NeuN expression. (**a**) Immunofluorescent staining in hippocampus probing for GFAP, a marker of astrocytes, and NeuN and (**b**) representation of fluorescence (corrected total fluorescence; CTF) of NeuN and GFAP represented as a % of wild-type. A total of 3 fields were imaged and mean values ± SEM are graphed. Nuclei were counterstained with DAPI. Micrographs were taken at 200× magnification. Scale bar = 100 microns.

**Figure 9 nutrients-17-02701-f009:**
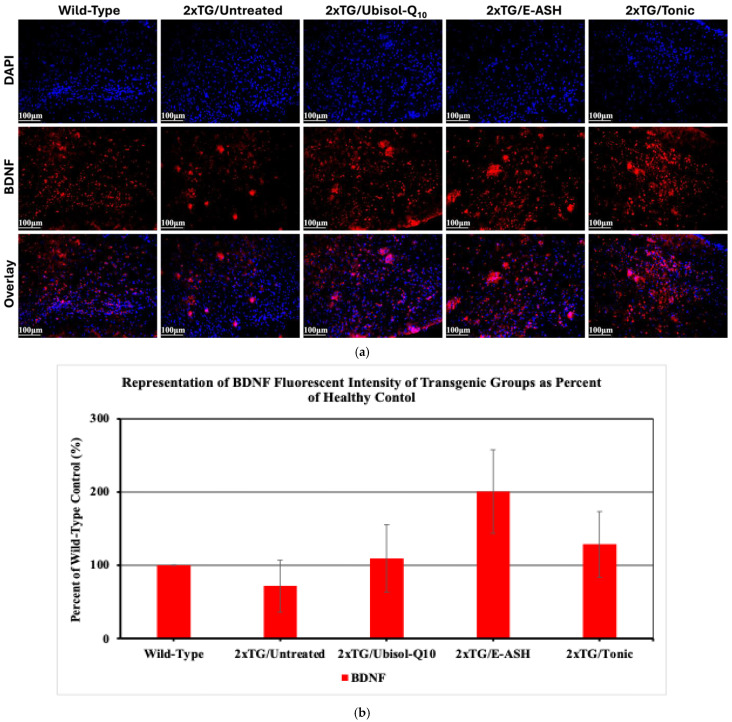
Effect of Ubisol-Q10 and E-ASH on BDNF expression. (**a**) Immunofluorescent staining in cortex probing for BDNF and (**b**) representation of fluorescence (corrected total fluorescence; CTF) of BDNF represented as a % of wild-type control. A total of 3 fields were imaged and mean values ± SEM are graphed. Nuclei were counterstained with DAPI. Micrographs were taken at 200× magnification. Scale bar = 100 microns.

**Figure 10 nutrients-17-02701-f010:**
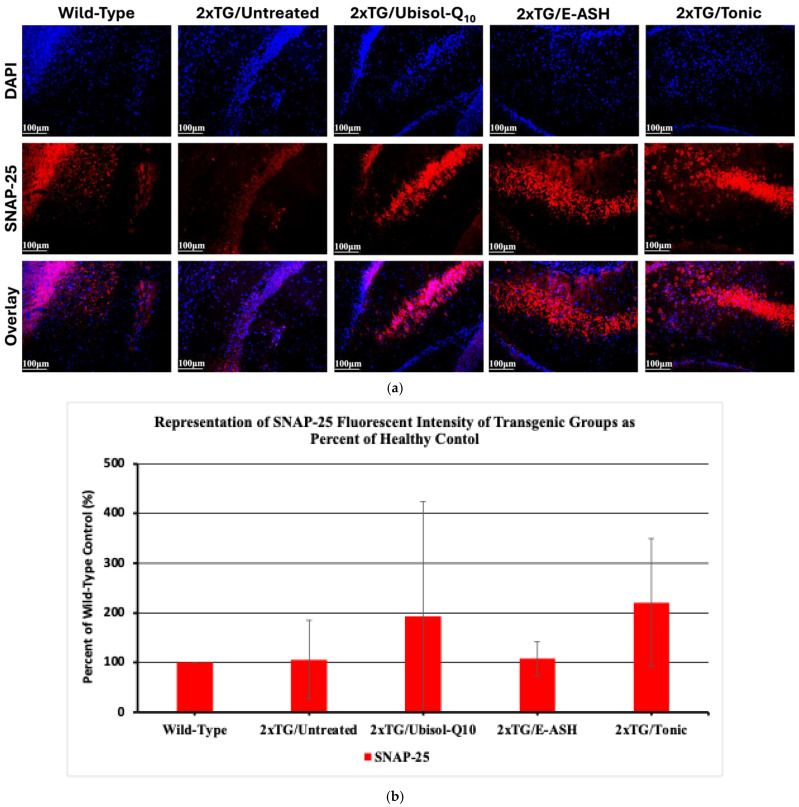
Effect of Ubisol-Q10 and E-ASH on synaptic health and density. (**a**) Immunofluorescent staining in hippocampal region probing for SNAP-25 and (**b**) representation of fluorescence (corrected total fluorescence; CTF) of SNAP-25 represented as a % of wild-type. A total of 3 fields were imaged and mean values ± SEM are graphed. Nuclei were counterstained with DAPI. Micrographs were taken at 200× magnification. Scale bar = 100 microns.

**Figure 11 nutrients-17-02701-f011:**
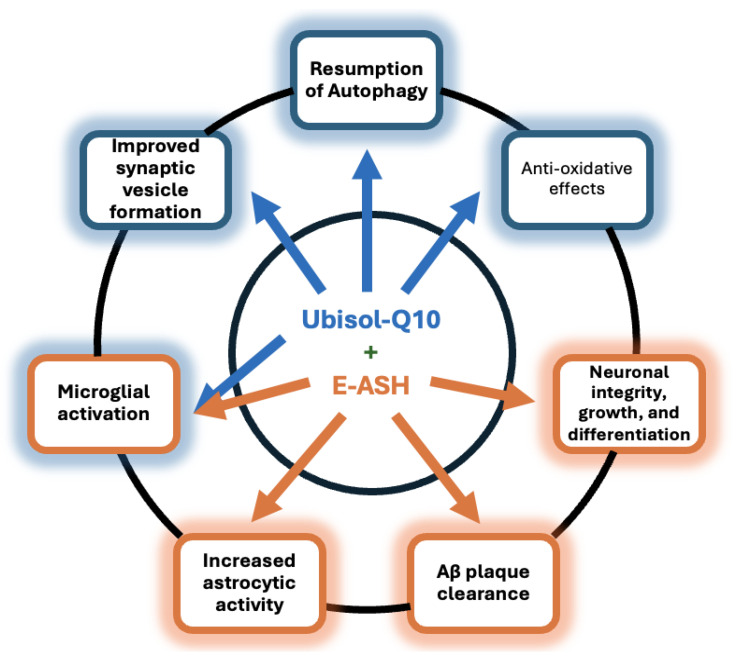
Representation of mechanistic highlights of Ubisol-Q10 and ethanolic Ashwagandha extract (E-ASH) based on the current results. Arrows indicates the bioactivity related to each treatment.

**Table 1 nutrients-17-02701-t001:** Withanolide and flavonoid content of E-ASH [56].

	Withanolides				Flavonoids
	Content (mg/mL)				Content (mg/100 mL)
	Withaferin A	12-Deoxy-Withastramonolide	Withanolide A	Withanolide B	Quercetin
Assay 1	13.0	3.7	5.3	1.8	Mean 1.63 ± 0.07
Assay 2	13.9	4.0	5.6	2.0	
Assay 3	13.8	3.8	5.6	1.7	
Assay 4	13.6	3.9	5.5	1.8	
Mean ± standard deviation	13.6 ± 0.4	3.8 ± 0.1	5.5 ± 0.1	1.9 ± 0.1	

## Data Availability

The data supporting the reports within the current study are available upon request. Raw data is not publicly available due to privacy and ethical reasons.

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
