# Peer review of "Investigation into Efficacy and Mechanisms of Neuroprotection of Ashwagandha Root Extract and Water-Soluble Coenzyme Q10 in a Transgenic Mouse Model of Alzheimer’s Disease"

_nutrients, 2025, doi:10.3390/nu17162701_

Round 1

Reviewer 1 Report

Comments and Suggestions for Authors

The manuscript “Investigation into Efficacy and Mechanisms of Neuroprotection 2 of Ashwagandha Root Extract and Water-Soluble Coen-3 zymeQ10 in a Transgenic Mouse Model of Alzheimer’s Disease” describes the neuroprotective effects of coenzyme Q10 and Ashwagandha extract in transgenic mouse models of Alzheimer's disease.

The authors presented as main objective to investigate the efficacy and neuroprotective mechanisms of action of coenzyme Q10 and Ashwagandha extract in the progression of Alzheimer's disease and addressed factors such as amyloid-β (Aβ) plaque deposition, oxidative stress, neuroglial inflammation, autophagy, synaptic function and (spatial and non-spatial memory, which is very interesting, since Alzheimer's disease is a disease with multiple pathogenic factors occurring at the same time. Also, the authors used a relevant animal model and presented a rigorous methodology, combining behavioral, histological, and biochemical assays.

I observed some limitations such as the clear correlation between astrogliosis and neuroprotection. Also, the animal studies do not always reflect human results and clinical trials are needed. In addition, the short observation period in behavioral tests may be limiting and longer periods could provide additional information about Alzheimer's disease.

Furthermore, the authors mention in Table 1 the phytochemical analysis of the Ashwagandha root extract; however, there is no correlation with the results presented. What is the relevance of this data?

Taking into account that the treatment was given by drinking water, how did the authors control the dose administered? The animals could have received different concentrations of coenzyme Q10 and Ashwagandha extract and, thus, exhibited different responses to the same treatment.

Author Response

Dear Editor:

Thank you for your email regarding our manuscript (ID: nutrients-3781243). We would like to thank the reviewers for their time, thorough review, and valuable comments and suggestions. We have gone through each of the comments from each reviewer and revised the manuscript as per their suggestions and comments to our best abilities. Indeed, the revised manuscript is improved significantly. We have also responded to each of the comments as indicated below and have indicated the revisions in yellow highlights in the revised manuscript. Please find the revised manuscript and we are hoping to hear from you soon.

Sincerely,

Siyaram Pandey

P.S. the Micillic Lda patent holders have revised their tradename for the products to be “Ubisol-Q10” instead of “PTS-COQ10,” and “Tonic,” instead of “CereQ10.”

Below are the responses to each reviewer’s comments:

Reviewer 1 - Comments and Suggestions for Authors:

The manuscript “Investigation into Efficacy and Mechanisms of Neuroprotection 2 of Ashwagandha Root Extract and Water-Soluble CoenzymeQ10 in a Transgenic Mouse Model of Alzheimer’s Disease” describes the neuroprotective effects of coenzyme Q10 and Ashwagandha extract in transgenic mouse models of Alzheimer's disease.

The authors presented as main objective to investigate the efficacy and neuroprotective mechanisms of action of coenzyme Q10 and Ashwagandha extract in the progression of Alzheimer's disease and addressed factors such as amyloid-β (Aβ) plaque deposition, oxidative stress, neuroglial inflammation, autophagy, synaptic function and (spatial and non-spatial memory, which is very interesting, since Alzheimer's disease is a disease with multiple pathogenic factors occurring at the same time. Also, the authors used a relevant animal model and presented a rigorous methodology, combining behavioral, histological, and biochemical assays.

I observed some limitations such as the clear correlation between astrogliosis and neuroprotection. Also, the animal studies do not always reflect human results and clinical trials are needed. In addition, the short observation period in behavioral tests may be limiting and longer periods could provide additional information about Alzheimer's disease.

Furthermore, the authors mention in Table 1 the phytochemical analysis of the Ashwagandha root extract; however, there is no correlation with the results presented. What is the relevance of this data?

Taking into account that the treatment was given by drinking water, how did the authors control the dose administered? The animals could have received different concentrations of coenzyme Q10 and Ashwagandha extract and, thus, exhibited different responses to the same treatment.

Response to Reviewer 1:

Thank you very much for your detailed suggestions of improvement to our manuscript, “Investigation into Efficacy and Mechanisms of Neuroprotection of Ashwagandha Root Extract and Water-Soluble CoenzymeQ10 in a Transgenic Mouse Model of Alzheimer’s Disease.” We have applied your comments in the following ways:

  1. The correlation between astrogliosis and neuroprotection has been acknowledged in lines 604 – 612 of the revised manuscript. This is a very interesting result, and your suggestion to emphasize this relationship is important.
  2. Your suggestion to address species differences between mice and human studies, as well as highlight the importance of clinical studies has been included in lines 714-730 of the revised manuscript. Thank you for this helpful suggestion.
  3. Our short behavioural observation period is, indeed, a limitation which we aim to extend in future studies. A brief statement had been included in the original manuscript (lines 709-710); however, we recognize this point should be given more emphasis, as per your suggestion. We have included more information on future directions of the study in lines 711-717 of the revised manuscript.
  4. Thank you very much for your thoughtful insight into our phytochemical analysis We have recognized that this information was not given the proper attention and interpretations as it should be. We have added information on the relevance of this information in the Discussion section in lines 508-528.
  5. Information on our dosing logistics and control of dose administration has been added to the Materials and Methods section of the revised manuscript. Please see lines 144-147 for this information. Thank you for your advice.

Reviewer 2 Report

Comments and Suggestions for Authors

Dear Authors,

Investigation into Efficacy and Mechanisms of Neuroprotection of Ashwagandha Root Extract and Water-Soluble CoenzymeQ10 in a Transgenic Mouse Model of Alzheimer’s Disease 

Is a very well executed project. Alzheimer's disease is a big burden on the society, age population and family. I enjoyed reading the 

invitro assays

invivo assays (Transgenic mouse) etc. 

It might be better to explain before mouse model studies, why the other methods were not utilized. Any 3D organoids studies?

What is the active principle of ashwangandha? an extract is great. Ashwangandha has medicinal value. But, what the main ingrediant? As the medication has to cross the BBB, it is essential to know the structures.

The antibody assays for development of Alzheimers, and progression are very relevant. 

Author Response

Response to Reviewers for Manuscript, “Investigation into Efficacy and Mechanisms of Neuroprotection of Ashwagandha Root Extract and Water-Soluble CoenzymeQ10 in a Transgenic Mouse Model of Alzheimer’s Disease”

Dear Editor:

Thank you for your email regarding our manuscript (ID: nutrients-3781243). We would like to thank the reviewers for their time, thorough review, and valuable comments and suggestions. We have gone through each of the comments from each reviewer and revised the manuscript as per their suggestions and comments to our best abilities. Indeed, the revised manuscript is improved significantly. We have also responded to each of the comments as indicated below and have indicated the revisions in yellow highlights in the revised manuscript. Please find the revised manuscript and we are hoping to hear from you soon.

Sincerely,

Siyaram Pandey

P.S. the Micillic Lda patent holders have revised their tradename for the products to be “Ubisol-Q10” instead of “PTS-COQ10,” and “Tonic,” instead of “CereQ10.”

Below are the responses to each reviewer’s comments:

Reviewer 2 - Comments and Suggestions for Authors

Dear Authors,

Investigation into Efficacy and Mechanisms of Neuroprotection of Ashwagandha Root Extract and Water-Soluble CoenzymeQ10 in a Transgenic Mouse Model of Alzheimer’s Disease 

Is a very well executed project. Alzheimer's disease is a big burden on the society, age population and family. I enjoyed reading the 

invitro assays

invivo assays (Transgenic mouse) etc. 

It might be better to explain before mouse model studies, why the other methods were not utilized. Any 3D organoids studies?

What is the active principle of ashwangandha? an extract is great. Ashwangandha has medicinal value. But, what the main ingrediant? As the medication has to cross the BBB, it is essential to know the structures.

The antibody assays for development of Alzheimers, and progression are very relevant. 

Response to Comments from Reviewer 2:

Thank you for your thoughtful insight into how we can improve our manuscript. We have taken your suggestions into consideration:

  1. Before we moved to animal studies, we had completed extensive tests to fibroblasts from AD patients, as well as neuroprotection in neuronal cells, to ensure the safety and efficacy of Ubisol-Q10. We have clearly indicated these works and highlighted it on the revised manuscript in lines 79-82. Thank you for your comment. I have included the published work, which is cited in the current manuscript, below for your reference:

McCarthy S, Somayajulu M, Sikorska M, et al. Paraquat induces oxidative stress and neuronal cell death; neuroprotection by water-soluble Coenzyme Q10. Toxicol Appl Pharmacol 2004; 201: 21-31. DOI: 10.1016/j.taap.2004.04.019.

Somayajulu M, McCarthy S, Hung M, et al. Role of mitochondria in neuronal cell death induced by oxidative stress; neuroprotection by Coenzyme Q10. Neurobiol Dis 2005; 18: 618-627. DOI: 10.1016/j.nbd.2004.10.021.

  1. Thank you for your suggestion to include more information on the relevance of our E-ASH composition. The active principles of our solution are important to acknowledge to give insight into the strengths and weaknesses of our solution. We have included more information on the active components of our E-ASH in lines 508-528.

Reviewer 3 Report

Comments and Suggestions for Authors

Dear Authors:

The manuscript presents a well-designed and comprehensive study evaluating the neuroprotective effects of water-soluble coenzyme Q10 (PTS-COQ10) and ethanolic Ashwagandha root extract (E-ASH) in a transgenic mouse model of Alzheimer’s disease (AD). The work is highly relevant given the pressing need for multi-targeted therapeutic approaches for AD.

Comments and suggestions:

  • Introduction (Lines 40-103): The introduction is well-structured and provides a comprehensive background on Alzheimer’s disease and the rationale for using multi-targeting natural health products.

Consider briefly stating in the final paragraph of the introduction how the present study builds upon prior work with PTS-COQ10 and Ashwagandha (e.g., Muthukumaran et al., 2018) to clearly frame the novelty of combining these compounds.

  • Methods (Lines 154–236): The description of the X-maze adaptation for NLR and NOR testing is clear. It may be helpful to include a schematic with labels for novel object vs. novel location changes to assist readers unfamiliar with this paradigm.

Since some video file corruption occurred, a short sentence in the limitations section noting the proportion of missing data and its potential effect on results would be appropriate.

  • Methods (Lines 270-308): Quantification is described as representing “general trends” rather than absolute values. Consider clarifying whether these relative intensity measures were statistically tested or are intended as descriptive only.

In the figures, ensure scale bars are legible and that fluorescence intensity graphs indicate the statistical significance levels for all group comparisons.

  • Results (Lines 310-378): In the NOR test, E-ASH and PTS-COQ10 groups decreased performance between phases, which contrasts with CereQ10. Consider adding a brief interpretation in the discussion as to why this pattern might occur (e.g., compound-specific time-dependent effects, plateau in performance).

  • Results (Lines 379-488): The finding that CereQ10 reduced both plaque size and microglial activation more than either treatment alone is compelling. It might be beneficial to quantify effect sizes (e.g., % reduction relative to untreated) directly in the figure legends for emphasis.

For the autophagy markers, the lack of effect with E-ASH is interesting-adding a comment in the discussion about possible alternative clearance pathways would enhance the mechanistic insight.

  • Discussion (Lines 489-670): The discussion effectively integrates the behavioral and biochemical findings. However, more emphasis could be placed on translational implications: how dosing in mice might compare to human-equivalent doses, known safety profiles, and possible delivery methods for clinical use.

The section on astrocyte activation (Lines 540-572) could briefly acknowledge that GFAP upregulation can also be a marker of reactive gliosis and distinguish why the authors interpret this as neuroprotective in the present study.

When discussing oxidative stress reduction (Lines 573-586), it would be useful to note whether these antioxidant effects could indirectly contribute to improved cognitive performance through preserved synaptic integrity.

  • Figures and Tables: Figures are generally clear and well-presented. Ensure all abbreviations (e.g., NC, NL, NO, CTF) are defined in figure legends for reader clarity.

Consider combining certain multi-panel figures (e.g., Figures 5-10) into a schematic summarizing the main biochemical effects of each treatment to visually reinforce the multi-targeting concept.

  • Conclusions (Lines 671–698): The conclusion is strong and well-supported by data. You might add a brief “Future Directions” sentence highlighting the need for mechanistic validation of autophagy involvement and long-term behavioral studies beyond 9 months in this model.

Kind regards,

Author Response

Response to Reviewers for Manuscript, “Investigation into Efficacy and Mechanisms of Neuroprotection of Ashwagandha Root Extract and Water-Soluble CoenzymeQ10 in a Transgenic Mouse Model of Alzheimer’s Disease”

Dear Editor:

Thank you for your email regarding our manuscript (ID: nutrients-3781243). We would like to thank the reviewers for their time, thorough review, and valuable comments and suggestions. We have gone through each of the comments from each reviewer and revised the manuscript as per their suggestions and comments to our best abilities. Indeed, the revised manuscript is improved significantly. We have also responded to each of the comments as indicated below and have indicated the revisions in yellow highlights in the revised manuscript. Please find the revised manuscript and we are hoping to hear from you soon.

Sincerely,

Siyaram Pandey

P.S. the Micillic Lda patent holders have revised their tradename for the products to be “Ubisol-Q10” instead of “PTS-COQ10,” and “Tonic,” instead of “CereQ10.”

Below are the responses to each reviewer’s comments:

Reviewer 3 - Comments and Suggestions for Authors

Dear Authors:

The manuscript presents a well-designed and comprehensive study evaluating the neuroprotective effects of water-soluble coenzyme Q10 (PTS-COQ10) and ethanolic Ashwagandha root extract (E-ASH) in a transgenic mouse model of Alzheimer’s disease (AD). The work is highly relevant given the pressing need for multi-targeted therapeutic approaches for AD.

Comments and suggestions:

  • Introduction (Lines 40-103): The introduction is well-structured and provides a comprehensive background on Alzheimer’s disease and the rationale for using multi-targeting natural health products.

Consider briefly stating in the final paragraph of the introduction how the present study builds upon prior work with PTS-COQ10 and Ashwagandha (e.g., Muthukumaran et al., 2018) to clearly frame the novelty of combining these compounds.

  • Methods (Lines 154–236): The description of the X-maze adaptation for NLR and NOR testing is clear. It may be helpful to include a schematic with labels for novel object vs. novel location changes to assist readers unfamiliar with this paradigm.

Since some video file corruption occurred, a short sentence in the limitations section noting the proportion of missing data and its potential effect on results would be appropriate.

  • Methods (Lines 270-308): Quantification is described as representing “general trends” rather than absolute values. Consider clarifying whether these relative intensity measures were statistically tested or are intended as descriptive only.

In the figures, ensure scale bars are legible and that fluorescence intensity graphs indicate the statistical significance levels for all group comparisons.

  • Results (Lines 310-378): In the NOR test, E-ASH and PTS-COQ10 groups decreased performance between phases, which contrasts with CereQ10. Consider adding a brief interpretation in the discussion as to why this pattern might occur (e.g., compound-specific time-dependent effects, plateau in performance).
  • Results (Lines 379-488): The finding that CereQ10 reduced both plaque size and microglial activation more than either treatment alone is compelling. It might be beneficial to quantify effect sizes (e.g., % reduction relative to untreated) directly in the figure legends for emphasis.

For the autophagy markers, the lack of effect with E-ASH is interesting-adding a comment in the discussion about possible alternative clearance pathways would enhance the mechanistic insight.

  • Discussion (Lines 489-670): The discussion effectively integrates the behavioral and biochemical findings. However, more emphasis could be placed on translational implications: how dosing in mice might compare to human-equivalent doses, known safety profiles, and possible delivery methods for clinical use.

The section on astrocyte activation (Lines 540-572) could briefly acknowledge that GFAP upregulation can also be a marker of reactive gliosis and distinguish why the authors interpret this as neuroprotective in the present study.

When discussing oxidative stress reduction (Lines 573-586), it would be useful to note whether these antioxidant effects could indirectly contribute to improved cognitive performance through preserved synaptic integrity.

  • Figures and Tables: Figures are generally clear and well-presented. Ensure all abbreviations (e.g., NC, NL, NO, CTF) are defined in figure legends for reader clarity.

Consider combining certain multi-panel figures (e.g., Figures 5-10) into a schematic summarizing the main biochemical effects of each treatment to visually reinforce the multi-targeting concept.

  • Conclusions (Lines 671–698): The conclusion is strong and well-supported by data. You might add a brief “Future Directions” sentence highlighting the need for mechanistic validation of autophagy involvement and long-term behavioral studies beyond 9 months in this model.

Kind regards,

Response to Reviewer 3:

Thank you for your thorough and insightful suggestions to our manuscript. Indeed, we find your suggestions are very helpful and have improved the quality of our manuscript in the following ways:

  1. Thank you for your suggestion to highlight the novelty of our work in the introduction. A brief statement has been added to lines 99-102.
  2. Figure 2 has been amended to add clarity to the NOR and NLR test objectives. Thank you for this recommendation.
  3. It is, indeed, important to highlight the limitations this data loss has caused to our data set. We have acknowledged your suggestion in lines 707-710 of the revised manuscript.
  4. Thank you for your suggestion to clarify the intention of our quantification of qualitative data. We did not conduct any statistical analyses on the histochemical data and have added an acknowledgment of this point in lines 313-314 of the revised manuscript.
  5. Your suggestion to enhance the visibility of our scale bars has been considered and the updated figures have been added to the revised manuscript. Thank you for your helpful recommendation. As for your suggestion to include significance levels for the group comparisons, we did not conduct statistics on the immunofluorescent data, and therefore, we cannot include your recommendation in our manuscript. Qualitative analysis was our main objective with this dataset. However, we will take your comment into great consideration in our future studies.
  6. Your insightful observation of the difference in performance over time between the treatment groups has been well received. We have included an interpretation of this phenomenon in lines 703-704 and lines 706-709 of the revised manuscript.
  7. Your recommendation to emphasize the drastic downregulation of both plaque-load and microgliosis is greatly appreciated. This result is, indeed, very interesting and we have acknowledged your suggestion by adding the quantified mean values of % relative to controls (untreated transgenic/wild-type) in the figure 5b and c legends.
  8. Your thorough interpretation of the autophagy markers, and the possible alternative plaque clearance pathways of E-ASH is greatly appreciated. This is an incredibly thought-provoking observation, which we have acknowledged in lines 551-556 of the revised manuscript.
  9. Your suggestion to highlight the mouse to human translational implications of our Ubisol-Q10 and E-ASH treatment has been considered and included in the revised manuscript in lines 716-730. This is an important point to make when evaluating novel treatments for human diseases, and we believe the enhanced emphasis of this point has greatly improved our relatability. Thank you for this comment.
  10. Your comment that there is a fine line between neuro-toxic and neuro-protective gliosis is very insightful and we have considered this interpretation in the revised manuscript in lines 583-593 and 604-612. Thank you for your perceptive comprehension of our results.
  11. The results for oxidative stress and the behavioural data and your thought-provoking connection between the two is greatly appreciated. This is a very interesting connection and deserves acknowledgement in the current manuscript. Your comment has been applied in lines 672-676.
  12. The abbreviations for NC, NL, NO, and CTF have been defined in the figure legend to improve clarity of these terms. We have also added the definitions directly into the figures (Fig 3 and 4) for good measure. Thank you for your helpful suggestion.
  13. Your comment that the impact of the results can be improved by schematically summarizing the treatments’ highlights is well appreciated. Your suggestion has been acknowledged by the addition of Figure 11. Thank you very much for this recommendation, and we believe the addition of a visual summary has improved the quality and readability of our manuscript.
  14. A brief “Future Directions” sentence has been added to the concluding section of the current manuscript. As per your helpful suggestion, we have stated our future aim of clarifying the autophagy-related roles of the treatments, as well as to lengthen our behavioural observation period. Thank you for your considerate suggestion.

Round 2

Reviewer 3 Report

Comments and Suggestions for Authors

Dear authors:

Thank you very much for addressing my comments and suggestions.

Kind regards,